# Characterization of the Ang/Tie2 Signaling Pathway in the Diaphragm Muscle of DMD Mice

**DOI:** 10.3390/biomedicines11082265

**Published:** 2023-08-14

**Authors:** Yiming Lin, Andrew McClennan, Lisa Hoffman

**Affiliations:** 1Department of Pathology and Laboratory Medicine, Western University, London, ON N6A 3K7, Canada; ylin382@uwo.ca; 2The Lawson Health Research Institute, London, ON N6C 2R5, Canada; amcclenn@uwo.ca; 3Department of Medical Biophysics, Western University, London, ON N6A 3K7, Canada

**Keywords:** Duchenne muscular dystrophy, angiopoietin, Tie2, angiogenesis, diaphragm

## Abstract

In Duchenne muscular dystrophy (DMD), angiogenesis appears to be attenuated. Local administration of angiopoietin 1 (Ang1) has been shown to reduce inflammation, ischemia, and fibrosis in DMD mice. Ang1 is a vital vascular stabilizing factor that activates the endothelial cell receptor Tie2, leading to downstream pro-survival PI3K/Akt pathway activation and eNOS phosphorylation. In this study, we aimed to characterize the Ang/Tie2 signaling pathway within the diaphragm muscle of mouse models of DMD. Utilizing ELISA, immunoblots, and RT-qPCR, we demonstrated that Ang1 was downregulated, while the antagonist angiopoietin 2 (Ang2) was upregulated, leading to a decreased Ang1/Ang2 ratio. This correlated with a reduction in the phosphorylated Tie2/total Tie2 ratio. Interestingly, no significant differences in Akt or eNOS phosphorylation were observed, although DMD murine models did have elevated total Akt protein concentrations. These observations suggest that Ang1/Tie2 signaling may be dysregulated in the diaphragm muscle of DMD and further investigations may lead to new therapeutic interventions for DMD.

## 1. Introduction

A significant obstacle faced by stem cell therapy and other regenerative therapies developed to treat Duchenne muscular dystrophy (DMD) is the inflamed and fibrotic microenvironment into which these therapies are introduced. This inflamed microenvironment is not favorable for tissue regeneration due to reduced oxygen and nutrient delivery to the tissues, which impairs the tissue’s endogenous repair process [1]. With reduced blood perfusion, the delivery of viral vectors and stem cells alike is severely hindered. This can lead to fatal complications in DMD patients when vital respiratory muscles, such as the diaphragm, are affected. Thus, the addition of vascular therapy to treat muscle ischemia and enhance endogenous muscle regeneration has been proposed [2,3,4,5].

The development of new vascular networks is initiated upon stimulation by several angiogenic factors such as vascular endothelial growth factor (VEGF) and hypoxia-inducible factor-1 (HIF-1α) [6]. Both VEGF and HIF-1α are prominent angiogenic factors expressed by satellite cells during muscle regeneration. Satellite cells isolated from 12-month-old *mdx* mice demonstrated reduced expression of both VEGF and HIF-1α, leading to reduced angiogenic capabilities in vitro [7]. Previous investigation has shown a significant reduction in VEGF and Ang1 concentration in the fibrotic diaphragm of *mdx/utrn^+/−^* mice, possibly resulting in deficiencies in both angiogenesis initiation and angiogenesis maturation that is unable to support endogenous muscle repair [8]. 

Induced VEGF overexpression has been shown to result in significant increases in endogenous muscle regeneration, evidenced by an increase in the number of myogenin-positive satellite cells and the number of regenerating myofibers [9]. However, while VEGF treatment appears promising, the sole usage of VEGF in correcting the vascular defects of DMD has been challenged. Our lab has previously shown that intramuscular administration of VEGF into the gastrocnemius muscle of a *mdx/utrn^+/−^* mouse does not improve blood perfusion, nor does it decrease collagen deposition. Interestingly, the administration of VEGF in combination with angiopoietin 1 (Ang1) improves blood volume and reduces fibrosis. Subsequent sole administration of Ang1 into gastrocnemius muscles of *mdx/utrn^+/−^* mice increases vascular density and vessel maturation and decreases collagen deposition [10]. 

The angiopoietin family of proteins consists of Ang1, angiopoietin 2 (Ang2), angiopoietin 3 (Ang3), and angiopoietin 4 (Ang4) [11]. Ang1 and Ang2 are the main ligands for Tie2, an endothelial cell-specific receptor tyrosine kinase. Tie2 undergoes specific subcellular localization upon angiopoietin activation. This localization helps differentiate the signaling outcomes between the quiescent and angiogenic endothelium [12]. In motile angiogenic endothelium, matrix-bound Ang1 binds to Tie2 and localizes it to cell–extracellular matrix contacts, and proceeds with cell migration and proliferation processes. In quiescent endothelium, Ang1/Tie2 signaling translocates Tie2 on confluent endothelial cells into clusters at the endothelial cell–endothelial cell junctions and induces pro-survival, endothelial tight junction stabilization, and anti-inflammatory signals. This is primarily achieved through downstream recruitment and activation of PI3K and subsequent Akt phosphorylation. Akt signaling promotes endothelial cell survival through the upregulation of pro-survival molecules, including phosphorylated endothelial nitric oxide synthase (eNOS) and survivin expression, while simultaneously suppressing apoptotic pathways such as caspase-9- and B-cell lymphoma 2 (BCL2)-associated agonist of cell death (BAD) [13,14].

Ang2 primarily functions as the competitive antagonist to Ang1, inhibiting Tie2 signaling. Ang2 is primarily expressed by endothelial cells and is transcriptionally induced in response to hypoxia, high glucose concentration, shear stress, tumor necrosis factor-alpha (TNFα), and VEGF. Within quiescent endothelial cells, Ang2 is transcriptionally repressed by the transcription factor Kruppel-like factor 2 (KLF2) [15]. Endogenous and overexpressed Ang2 is stored within specialized endothelial storage granules called Weibel–Palade bodies (WPB). In homeostatic endothelium, the Ang1/Ang2 ratio heavily favors Ang1 due to its constitutive expression, while Ang2 is downregulated. This downregulation allows Ang1/Tie2 signaling to mediate vascular quiescence. However, upon endothelial activation, the secreted Ang2 shifts the Ang1/Ang2 ratio towards Ang2 [16]. 

Thus, we aimed to characterize Ang/Tie2 signaling in DMD to address the potential vascular and microenvironment defects in the diaphragm. This characterization may enable future curative treatments to fully realize their therapeutic potential when addressing the respiratory complications of DMD.

## 2. Materials and Methods

### 2.1. Animal Care and Genotyping

All animal protocols were conducted in strict accordance with the Canadian Council on Animal Care (CCAC) and were approved by the Animal Care Committee (Western University, London, ON, Canada). All experiments were performed at The Lawson Health Research Institute at St. Joseph’s Health Care (SJHC) in London, Ontario. C57BL/10ScSn mice (Jax Laboratories, cat#000476, Bar harbor, Me, USA), *mdx/utrn^+/−^* (Jax Laboratories, Bar harbor, Me, USA), were purchased from the Jackson Laboratory (Bar Harbor, ME, USA) and maintained at the Animal Care Facility at SJHC. Colonies were maintained under controlled conditions (19–23 °C, 12 h light/dark cycles) and allowed water and food *ad libitum*. Mice from two age ranges, either 8–10 weeks old or 15–20 weeks old, were used. Wild-type (WT) and *mdx*/*utrn*^+/*−*^ mice (*n* = 3–8 for all groups) of either sex were included in this study to represent varying levels of disease severity, ranging from healthy to mild and moderate disease severity. This approach helps account for the differences in disease severity between mouse models and human patients.

Genotyping was conducted using tail snip or ear notch tissues by polymerase chain reaction (PCR) with platinum *Taq* polymerase (Thermofisher, Waltham, MA, USA), using the following set of utrophin gene primers (Sigma): 5′-TGCAGTGTCTCCAATAAGGTATGAAC-3′, 5′-TGCCAAGTTCTAATTCCATCAGAAGCTG -3′ (forward primers) and 5′-CTGAGTCAAACAGCTTGGAAGCCTCC-3′ (reverse primer). Gel electrophoresis was performed to determine the molecular weight of the amplified DNA. 

### 2.2. Tissue Preparation

Mice were sacrificed via gas euthanasia followed by cervical dislocation upon reaching the desired age of 8–10 weeks or 15–20 weeks. Diaphragms used for hematoxylin and eosin (H&E) and Masson’s trichrome staining were immediately dissected, fixed in 10% formalin for 24–48 h, and embedded in paraffin. Section slides were produced from 5 µm thick slices taken from every 5th serial section. Diaphragm and lung tissues (used as positive control) used for western blot, enzyme-linked immunosorbent assay (ELISA), and qPCR analysis were dissected, immediately flash frozen in dry ice, and stored at −80 °C. 

### 2.3. Microscopy and Image Analysis

Tissue slides were sent to the Molecular Pathology department at Robarts Research Institute (London, ON, Canada) for H&E and Masson’s trichrome staining. Histological images were captured on a Zeiss Axioskop 50Fluorescence ERGO Trinoc microscope with a 20× objective, utilizing Northern Eclipse Image software (Empix Imaging Inc., Version 8.0, Mississauga, ON, Canada). A minimum of five non-overlapping images were taken of each section. Qualitative analysis assessing the presence of centrally nucleated myofibers as a sign of recently regenerated myofibers and inflammatory infiltrate (H&E) and collagen (Masson’s trichrome) as indicators of inflammation and fibrosis in the tissue.

### 2.4. RNA Extraction and cDNA Preparation

Frozen diaphragms were cut and weighed, yielding 10–20 mg of tissue. Samples underwent three freeze–thaw cycles using liquid nitrogen, then homogenized in Trizol Reagent (Ambion, Waltham, MA, USA)) using polypropylene pestles (Fisher Scientific, Hampton, NH, USA) and an Eppendorf tube. RNA purification was performed using a Direct-zol^TM^ RNA miniprep kit (Zymo, Irvine, CA, USA) according to the manufacturer’s instructions. Post-purification was performed through an in-solution Dnase I treatment using <10 µg of RNA sample, Dnase I (Zymo, Irvine, CA, USA), DNA digestion buffer (Zymo, Irvine, CA, USA), and water and incubated at room temperature for 15 min. Three volumes of Trizol to one volume of the treated sample were added. The RNA purification step using the Direct-zol^TM^ RNA miniprep kit was then repeated, and the RNA was suspended in 40 µL of RNAase-free water. RNA concentration and quality were quantified using a DeNovix DS-11 spectrophotometer. All samples were verified to have a 260/280 ratio above 1.9 and a 260/230 ratio between 2.0–2.2. 

cDNA synthesis was performed using 1 µg of total RNA with the High-Capacity cDNA Reverse Transcription Kit (Applied Biosystems, Waltham, MA, USA) following the manufacturer’s instructions. 

### 2.5. RT-qPCR

TaqMan Gene Expression Assays (Thermofisher, Waltham, MA, USA) were used to assay *Tek* (Tie2) (Mm004432343_m1), *Angpt1* (Mm00456503_m1), *Angpt2* (Mm00545822_m1) [17], *ActB* (Mm02619580_g1), *AP3D1* (Mm00475961_m1), and *GusB* (Mm01197698_m1). qPCR products were generated using the QuantStudio^TM^ 5 system with TaqMan Fast Advanced Master Mix (Applied Biosystems, Waltham, MA, USA). Quantification was performed using Design & Analysis Software v2.4.3 (Thermofisher, Waltham, MA, USA). Expression of *Tek*, *Angpt1*, and *Angpt2* was normalized to the geometric mean of control genes (*ActB*, *AP3D1*, and *GusB*) validated for use in mdx mice [18].

### 2.6. Protein Extraction

Tissues were weighed and then homogenized in lysis buffer (20 mM Tris (pH 7.8), 17 mM NaCl, 2.7 mM KCl, 1 mM MgCl_2_, 1% Triton X-100, 10% (*w*/*v*) glycerol, 1 mM EDTA, supplemented with protease inhibitor (Roche, Mannheim, Germany) and phosphatase inhibitor (Abcam, Cambridge, UK)). The ratio of tissue weight (mg) to lysis buffer volume (µL) was maintained at 1:10. Homogenization was performed by incubating the samples for 4 h at 4 °C with continuous agitation. After, samples were centrifuged at 13,000 rpm for 15 min, and the supernatants were collected. Total protein quantification was performed using the Bicinchoninic acid assay (Pierce, Waltham, MA, USA). All samples were run in duplicates, and absorbance was measured at 575 nm with an iMark^TM^ Microplate Absorbance Reader. 

### 2.7. Western Blot

An amount of 40 µg of protein was combined in a 1:1 ratio with Laemmli sample buffer with 2-mercaptoethanol, heat denatured at 70 °C for 10 min, and loaded onto mini-PROTEAN TGX Stain-Free^TM^ Precast Gels (Bio-Rad, Hercules, CA, USA). An amount of 15 µL of PageRuler Plus Prestained Protein ladder (Thermofisher, Waltham, MA, USA) was loaded onto the gel. Protein separation via gel electrophoresis was run at 80 V for 30 min, followed by 60 min at 120 V. The gel was cut and activated using UV light with the Bio-Rad Gel Doc system (Bio-Rad, Hercules, CA, USA). Protein was transferred onto a PVDF membrane using the Transblot Turbo machine (Bio-Rad, Hercules, CA, USA) running for 10 min at a constant current of 1.3 A, up to 25 V. Membranes were blocked with either 5% bovine serum albumin (BSA) in tris-buffered saline containing 0.1% tween 20 (TBS-T) or 5% skim milk in 0.1% TBS-T for 1 h with continuous agitation. After blocking, membranes were incubated with primary anti-Ang1 (Abcam, ab8451, 1:1000, Cambridge, United Kingdom), Ang2 (Invitrogen, PA5–27297, 1:1000, Waltham, MA, USA), AKT (Cell Signaling, #9272, 1:2000, Danvers, MA, USA), pAKT ser473 (Cell Signaling, #9271S, 1:500, Danvers, MA, USA), eNOS (Abcam, ab76198, 1:500, Cambridge, United Kingdom), or Phospho-eNOS ser1177 (MyBioSource, MBS9601018, 1:1000, San Diego, CA, USA) antibodies diluted in either 5% BSA TBS-T or 5% skim milk in 5% TBS-T at 4 °C overnight. Afterward, the membranes were incubated with an anti-rabbit HRP secondary antibody (Abcam, ab6721, 1:5000, Cambridge, UK) or anti-mouse HRP secondary antibody (Abcam, ab97023, 1:5000, Cambridge, UK) for 1 h. Membranes were again thoroughly washed three times for five minutes with TBS-T. Total protein was visualized and imaged on the Bio-Rad Gel Doc system. For chemiluminescent detection, the membranes were incubated with SuperSignal West Pico PLUS chemiluminescent substrate (ThermoFisher, Waltham, MA, USA) for one minute. Bands were visualized using chemiluminescence using ImageLab (Bio-Rad, version 6.1.0, Hercules, CA, USA) on the Bio-Rad Gel Doc. The signal was normalized to the total protein signal from the stain-free blots [19].

### 2.8. Enzyme-Linked Immunosorbent Assay

To quantify levels of Ang1, Ang2, Tie2, and pTie2 in diaphragm lysates, the following kits were used: Mouse Angiopoietin 1 Elisa Kit (Develop, Pottstown, PA, USA), Quantikine Mouse Angiopoietin-2 (R&D Systems, Minneapolis, MN, USA), Quantikine Mouse Tie2 (R&D Systems, Minneapolis, MN, USA), Mouse Phosphorylated Tie 2 Duoset IC Elisa Kit (R&D Systems, Minneapolis, MN, USA), and Mouse VE-PTP Elisa Kit (Blue gene, Pottstown, PA, USA). Samples underwent ½ dilutions, and all samples were run in duplicates. Each kit was performed according to the manufacturer’s recommended instructions. The absorbance of the samples was measured with an iMark^TM^ Microplate Absorbance Reader at 450 nm and subtracted by absorbance at 570 nm. A standard curve was generated using the mean absorbance and concentration of each standard in GraphPad Prism for Windows (GraphPad Software, version 8.00, La Jolla, CA, USA).

### 2.9. Statistical Analysis

An ordinary two-way ANOVA followed by Tukey’s honest significance test was performed to compare the difference between groups using GraphPad Prism for Windows (GraphPad Software, version 8.00, La Jolla, CA, USA). Differences between groups were considered significant at a *p*-value of <0.05. Power calculations were conducted to determine the appropriate sample size for the experiments.

## 3. Results

### 3.1. Ang1/Ang2 ratio Is Skewed towards Ang2 in mdx/utrn^+/−^ Mice Relative to Age-Matched Wild-Type Mice

Utilizing Rt-qPCR to evaluate *Angpt1* and *Angpt2* mRNA expression, young (8–10 weeks old) *mdx/utrn^+/−^* mice showed significantly lower relative *Angpt1* expression compared to young WT (*p* = 0.0022) (Figure 1A). No significant differences in *Angpt2* mRNA expression were observed between the different genotypes of the young mice (Figure 1A). Interestingly, the subsequent protein data did not reflect the mRNA data. Using ELISA to quantify protein concentrations, young *mdx/utrn^+/−^* mice (1829.437 ± 1369.454 pg/mL) did not show significantly lower Ang1 protein concentration compared to young WT mice (2774.798 ± 1908.568) (*p* = 0.3075) (Figure 2A). In contrast, young *mdx/utrn^+/−^* mice (1226.882 ± 658.067 pg/mL) (*p* = 0.0123) had significantly higher Ang2 protein concentrations compared to young WT mice (346.888 ± 137.783 pg/mL) (Figure 2B).

In mature (15–20 weeks) *mdx/utrn^+/−^* mice, both *Angpt1* and *Angpt2* mRNA expression were significantly lower relative than that of mature WT mice (Figure 1A). This was supported by the subsequent ELISA protein concentration analysis, which demonstrated a significant decrease in Ang1 protein concentration in *mdx/utrn^+/^^−^* mice (965.069 ± 473.061 pg/mL) (*p* = 0.0042) compared to mature WT mice (3163.722 ± 713.035) (Figure 2A). However, mature *mdx/utrn^+/^^−^* mice did not show a significantly lower Ang2 protein concentration (1493.863 ± 325.298 pg/mL) (*p* = 0.4078) compared to the mature WT mice (1101.879 ± 375.764 pg/mL) (Figure 2B). 

Upon analysis of relative fold expression ratios between *Angpt1* and *Angpt2* mRNA, both young and mature *mdx/utrn^+/−^* mice had significantly lower *Angpt1*/*Angpt2* relative mRNA expression ratios compared to their age-matched WT counterparts (Figure 1B). Ang1/Ang2 protein ratio quantification supported the mRNA data, revealing a significantly lower Ang1/Ang2 ratio in both young (*p* < 0.0001) and mature (*p* < 0.0044) *mdx/utrn^+/−^* mice compared to their WT counterparts (Figure 2C). Two-way ANOVA did not reveal any statistically significant interaction between the effects of strain and age for *Angpt1*/*Angpt2* relative mRNA expression ratio (F (1, 9) = 0.02923, *p* = 0.8680) nor for Ang1/Ang2 protein ratio (F (1, 19) = 1.308, *p* = 0.2670). Altogether, these findings indicate a highly skewed Ang1/Ang2 ratio in favor of Ang2 signaling in the skeletal muscle of *mdx/utrn^+/^^−^* mice, which may result in Tie2 inactivation and downstream inhibition. 

### 3.2. pTie2/Tie2 Ratio Is Decreased in Mature mdx/utrn^+/−^ Mice Compared to Age-Matched Wild-Type Mice

Tek expression in young *mdx/utrn^+/−^* mice was not significantly different relative to young WT mice (*p* = 0.093) (Figure 3A). ELISA protein concentration analysis revealed elevated Tie2 expression in young *mdx/utrn^+/−^* mice (4542.796 ± 613.177 pg/mL), although it was not significantly higher than WT (3612.270 ± 238.670 pg/mL). Tie2 mRNA expression in mature 15–20-week-old mice similarly did not significantly differ between each genotype (*p* = 0.1665). Tie2 protein concentrations in mature *mdx/utrn^+/−^* mice (3230.151 ± 1449.219 pg/mL) similarly were not significantly different from those in mature WT mice (2261.032 ± 1646.605 pg/mL) (*p* = 0.6999) (Figure 3B). 

Phosphorylated Tie2 (pTie2) protein analysis via ELISA was then used to evaluate levels of Tie2 activation, which revealed no difference in pTie2 protein concentration in young *mdx/utrn^+/−^* mice (1078.286 ± 388.538 pg/mL) compared to young WT mice (1597.999 ± 532.224 pg/mL) (Figure 3C). No significant differences in pTie2 protein concentrations were detected between mature *mdx/utrn^+/−^* (908.122 ± 477.387 pg/mL) and WT mice (990.652 ± 663.280 pg/mL) (Figure 3C). 

Interestingly, the quantification of the ratio between pTie2 protein concentration and total Tie2 protein concentration showed a significant decrease in pTie2/Tie2 ratio in young *mdx/utrn^+/^^−^* mice (0.235 ± 0.065 pTie2/Tie2) and young WT mice (0.437 ± 0.119 pTie2/Tie2) (*p* = 0.0383) (Figure 3D). Similarly, this significant difference was observed between mature *mdx/utrn^+/−^* mice (0.259 ± 0.066 pTie2/Tie2) compared to mature WT mice (0.498 ± 0.14 pTie2/Tie2) (*p* = 0.0364) (Figure 3D). Two-way ANOVA revealed that there were no statistically significant interactions between the effects of strain and age for pTie2 (F (1, 8) = 0.5203, *p* = 0. 4913), Tie2 (F (1, 8) = 0.0008520, *p* = 0.9774), and the pTie2/Tie2 ratio (F (1, 8) = 0.1536, *p* = 0.7054). 

To determine whether the changes in pTie2/Tie2 ratio may be due to changes in the Tie2 regulatory protein VE-PTP^17^, ELISA was performed to determine VE-PTP protein concentration. No significant differences in VE-PTP protein concentrations were seen between young *mdx/utrn^+/−^* mice (2095.981 ± 625.203 pg/mL) compared to young WT mice (1888.470 ± 719.714 pg/mL) (*p* = 0.8486) (Figure 3E). Similarly, no significant differences in VE-PTP protein concentrations were seen between mature *mdx/utrn^+/−^* mice (2995 ± 1870.149 pg/mL) compared to mature WT mice (3252.685 ± 1255.981 pg/mL) (*p* = 0.2124). Taken together, this data showed that severely diseased mature mice had a lower pTie2/Tie2 ratio compared to WT mice. 

### 3.3. No Significant Differences in eNOS Phosphorylation or Akt Phosphorylation in mdx/utrn^+/−^ Mice

Endothelium homeostasis is partially preserved through Tie2-mediated Akt phosphorylation and downstrea eNOS phosphorylation [20]. Therefore, we aimed to assess whether the change in pTie2/Tie2 ratio influenced Akt phosphorylation and subsequent eNOS phosphorylation in the diaphragm of 8–10-week-old and 15–20-week-old WT and *mdx/utrn^+/^^−^* mice. Semi-quantitative western blot analysis, normalized to total protein, was used to quantify Akt and eNOS phosphorylation (Figure 4A and Figure 5A). Two-way ANOVA revealed no statistically significant interactions between the effects of strain and age for phosphorylated eNOS (F (1, 8) = 0.03012, *p* = 0. 7499) and phosphorylated Akt (F (1, 8) = 0.2679, *p* = 0.6187). In young 8–10-week-old mice, phosphorylated Akt protein levels were not significantly different between young *mdx/utrn^+/−^* mice and young WT mice (Figure 4B). Phosphorylated eNOS levels did not differ significantly between mature 15–20-week-old *mdx/utrn^+/−^* and WT mice (Figure 4B). The lack of difference in phosphorylated Akt levels was consistent with similarly unremarkable phosphorylated eNOS levels observed. Phosphorylated eNOS levels remained relatively similar between the disease model and WT in both young and mature mice (Figure 5B).

### 3.4. Total Akt Is Upregulated in Mature mdx/utrn^+/−^ Mice

Although no changes in eNOS phosphorylation nor Akt phosphorylation were observed, total eNOS and Akt protein levels were also evaluated to rule out any changes in endogenous protein concentration as a cause for observed results. Western blot analysis was performed using the same lysates as the eNOS and Akt phosphorylation experiments (Figure 4A and Figure 5A). Two-way ANOVA revealed no statistically significant interactions between the effects of strain and age for eNOS (F (1, 8) = 0.03004, *p* = 0. 8667) and Akt (F (1, 8) = 0.02585, *p* = 0.8763). eNOS levels in the disease model were not significantly different from the WT in either young or mature mice. Akt protein levels showed a significant increase in both young and mature *mdx/utrn^+/^^−^* mice (Figure 5B). This observation may suggest a decreased overall pAkt/Akt ratio in mature *mdx/utrn^+/^^−^* mice.

## 4. Discussion

The importance of Tie2 signaling via Ang1 and Ang2 is perhaps an overlooked area of research in DMD because of the insufficient characterization data of this signaling pathway in muscular dystrophies. To the best of our knowledge, this study is the first to examine the Ang1/Ang2 ratio in the diaphragm of DMD. The results indicate that the Ang1/Ang2 protein ratios in the diaphragm of DMD mice were significantly lower compared to age-matched WT controls, possibly due to decreased Ang1 and increased Ang2 levels. Separately, no significant differences in Tie2 expression nor Tie2 phosphorylation were found. However, the analysis of pTie2/Tie2 ratios revealed that fibrotic *mdx/utrn^+/−^* mice had a significantly lower pTie/Tie2 ratio relative to WT mice. Lastly, while Akt and eNOS phosphorylation did not show significant differences between the genotypes, total Akt protein levels were elevated in the diaphragm of mature *mdx/utrn*^+/*−*^ mice (Figure 4B).

Downregulation of Ang1 is required for an effective, acute inflammatory response due to Ang1′s anti-inflammatory and vascular stability effects. However, within the chronically inflamed microenvironment of the dystrophic diaphragm, Ang1 expression may be insufficient, leading to a persistently destabilized vasculature. Evidence has demonstrated that the antileakage action of Ang1 in the context of chronic inflammation depends on the presence of pericytes [21]. Ang1 also plays a crucial role in mediating myogenesis during muscle regeneration. Injection of cartilage oligomeric matrix protein (COMP) Ang1 variant, COMP-Ang1, a potent derivative of native Ang1, has been shown to stimulate muscle fiber regeneration through N-cadherin-mediated myogenin expression in myoblasts and rescue ischemic muscle injury [22]. Satellite cells, which are important for muscle regeneration, also express Ang1 [23]. Satellite cells lacking dystrophin are severely hindered in their ability to self-renew due to impairments in asymmetrical cell division [24,25]. The decrease in Ang1 expression observed in this study may further contribute to a diminished satellite cell population by reducing Ang1-mediated satellite cell self-renewal. Ang1 deficiency presented in this study may be indirect evidence for a reduction in both satellite cells and pericyte populations, which are associated with cycles of tissue regeneration and degeneration within the dystrophic skeletal muscles of DMD. 

The increase in Ang2 protein concentration seen within *mdx/utrn*^+/−^ mice is consistent with a proteome profiling analysis conducted on young 4–10-year-old human patients with DMD, which also reported elevated Ang2 protein levels in patient serums [26]. Ang2 is known to be capable of directly mediating endothelial cell differentiation and promoting sprouting angiogenesis in the presence of VEGF [27]. In the absence of VEGF, however, Ang2 induces endothelial cell apoptosis and vessel regression [28]. VEGF expression is downregulated in dystrophic satellite cells and dystrophic mice [7,10]. Therefore, a prolonged increase in Ang2 concentration may contribute to the detrimental vascular abnormalities observed in DMD, such as vessel regression, rather than promoting regeneration and inducing angiogenesis. 

The decrease in the Ang1/Ang2 ratio and subsequent inhibition of Ang1 signaling is required for the endothelium to transition from its quiescent state to an active state capable of effectively responding to inflammatory and angiogenic signals. However, a chronically decreased Ang1/Ang2 ratio from persistent Ang2 signaling results in chronically leaky vasculature that cannot support functional perfusion, leading to tissue ischemia. This may explain why the previous administration of VEGF alone into the similarly fibrotic gastrocnemius tissue was unable to induce mature vessel formation. Subsequent administration of Ang1 might have acted to shift the Ang1/Ang2 ratio towards Ang1 and allow for Ang1-mediated Tie2 activation and the conclusion of the angiogenic cascade. 

While Tie2 mRNA is upregulated during wound healing, studies have reported reduced Tie2 expression in conditions associated with severe vascular complications [29]. Upon binding and activation by Ang1 or Ang2, Tie2 is known to release the ligand, undergo internalization, and subsequently be targeted for degradation. Thus, it is possible that despite the reduction in Tie2 mRNA expression, Tie2 protein concentration remains elevated due to the high surface presentation of Tie2 in response to its inhibition by Ang2. Another factor may be the reduction of VEGF in DMD. Previous studies have shown that VEGF-induced sprouting angiogenesis is capable of inducing Tie2 cleavage and downregulating Tie2 mRNA expression in angiogenic endothelial cells [30]. The decrease in Tie2 mRNA expression may be correlated with the level of sprouting angiogenesis occurring within the tissue of *mdx/utrn*^+/−^ mice as endogenous muscle repair attempts. Although quantification of pTie2 in young mice did not reveal a significant decrease in phosphorylated protein concentration within the *mdx/utrn*^+/−^ mice, it did not correlate with the significant decrease in Ang1/Ang2 ratio shown previously. It is then possible that at 8–10 weeks of age, Tie2 phosphorylation, vascular stability, and angiogenic capabilities in *mdx/utrn*^+/−^ mice are conserved. As the mice aged, a significant trend towards decreased pTie2/Tie2 ratios was observed in *mdx/utrn*^+/−^ mice compared to WT mice, which did correlate with the changes in Ang1/Ang2 ratios. This may be a sign that vascular instability and angiogenic dysfunction proceed only after age or threshold. Thus, future experiments investigating the vascular activity of DMD mouse models should consider the age of the mice accordingly. 

One major target selected for analysis was Akt. PI3K-mediated Akt phosphorylation is a major downstream target of Tie2 and is known to contribute to cell survival, anti-inflammation, and vascular maturation [13,14]. Elevated Akt levels are associated with physiological muscle regeneration, and previous studies have reported several-fold increases in both pAkt and total Akt protein concentrations in the skeletal muscle of mdx and *mdx/utrn*^−/−^ mice, as well as biopsies from young patients with DMD [31]. However, a discrepancy between pAkt and total Akt levels has also been observed [31,32]. It has thus been suggested that Akt activation in DMD may be somewhat defective. Akt signaling is vital as it serves to induce an anti-apoptotic effect, promotes VEGF expression, and regulates endothelial cell migration. Indeed, inhibition of Akt signaling severely attenuates ischemia-mediated blood flow recovery and angiogenesis [33]. As such, the apparent decrease in overall Akt phosphorylation may be a sign of reduced vascular stability and reduced angiogenesis.

One major target of Akt downstream signaling within endothelial cells is Akt-mediated expression, phosphorylation, and NO production of eNOS. Previous studies have examined eNOS expression in canine models of DMD and found reduced eNOS expression and reduced cyclic guanosine monophosphate concentrations, leading to increased vascular leakiness [34]. The observed vascular defect can be directly resolved through the upregulation of eNOS. Elevations in Ang2 concentrations in sepsis are inversely proportional to NO-mediated microvascular reactivity [35]. Interestingly, our results did not reveal any significant differences in either eNOS expression or eNOS phosphorylation between the DMD and WT murine models. 

A major limitation of the current study is the sole focus on the diaphragm. The diaphragm is a mixed twitch skeletal muscle; therefore, the findings in this study cannot be directly extrapolated to all other skeletal muscles in the body. Future experiments should include the evaluation of other skeletal tissues, such as the tibialis anterior or extensor digitorum longus, to evaluate changes in the Ang/Tie2 ratios in fast twitch muscles. Furthermore, future experiments should evaluate Tie2, Ang1, and Ang2 expression specifically in endothelial cells and pericyte populations to determine the sources of the changes in the Ang1/Ang2 and pTie2/Tie2 ratios observed in this study. Additionally, there is an emerging hypothesis of pericyte deficiencies within DMD. Decreased Ang1 concentrations can result in pericyte detachment and pericyte–myofibroblast differentiation. Thus, future experiments should involve evaluating pericyte coverage in dystrophic tissues using techniques such as fluorescence-activated cell sorting or high-resolution fluorescence microscopy, including confocal and multiphoton microscopy. 

Taken together, characterization of the Ang1/Ang2 ratio and Tie2 activation opens another avenue of vascular therapy for DMD. Our laboratory has demonstrated that increasing Ang1 concentration and possibly increasing the Ang1/Ang2 ratio may have beneficial effects in promoting mature vasculature formation. It may also be worthwhile to investigate reducing free Ang2 levels in DMD as, in this context, it may shift the Ang1/Ang2 balance back towards an Ang1-favored ratio, which can promote Tie2 activation, maturation of the angiogenic cascade and establish a stabilized endothelium that can support functional perfusion and endogenous muscle repair. 

## Figures and Tables

**Figure 1 biomedicines-11-02265-f001:**
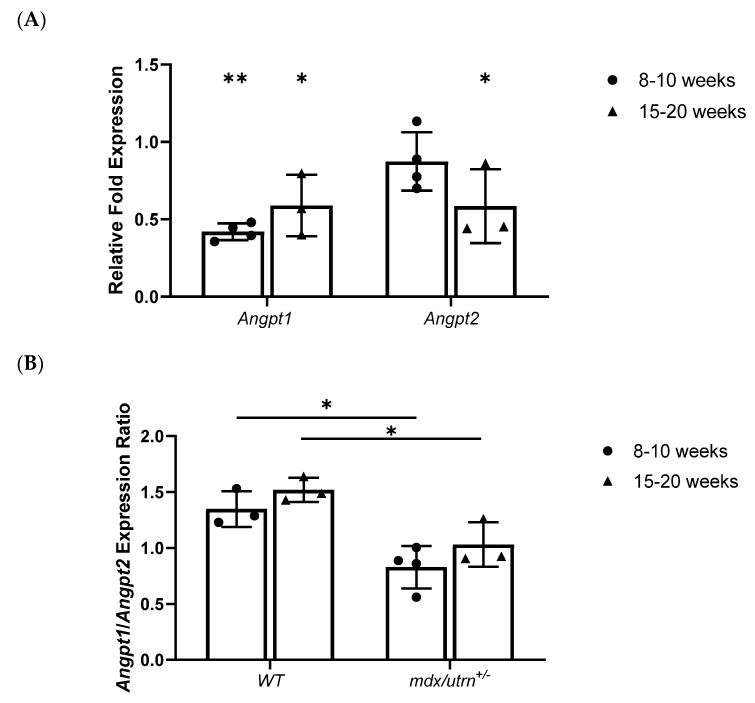
Rt-qPCR analysis of *Angpt1* and *Angpt2* gene expression. (**A**) RT-qPCR analysis of *Angpt1* and *Angpt2* relative fold expression in young (8–10 weeks old) and mature (15–20 weeks old) *mdx/utrn^+/−^* mice (*n* = 3–5) normalized to WT age-matched mice expression; reference genes: *ActB*, *AP3D1*, and *GusB*. (**B**) Quantification of the ratio of *Angpt1* expression relative to *Angpt2* expression diaphragm lysates of young and mature WT and *mdx/utrn^+/−^* mice (*n* = 3–4). Data are presented as individual values and mean ± SD. Statistics are performed using two-way ANOVA followed by Tukey’s test. * *p* < 0.05, ** *p* < 0.01.

**Figure 2 biomedicines-11-02265-f002:**
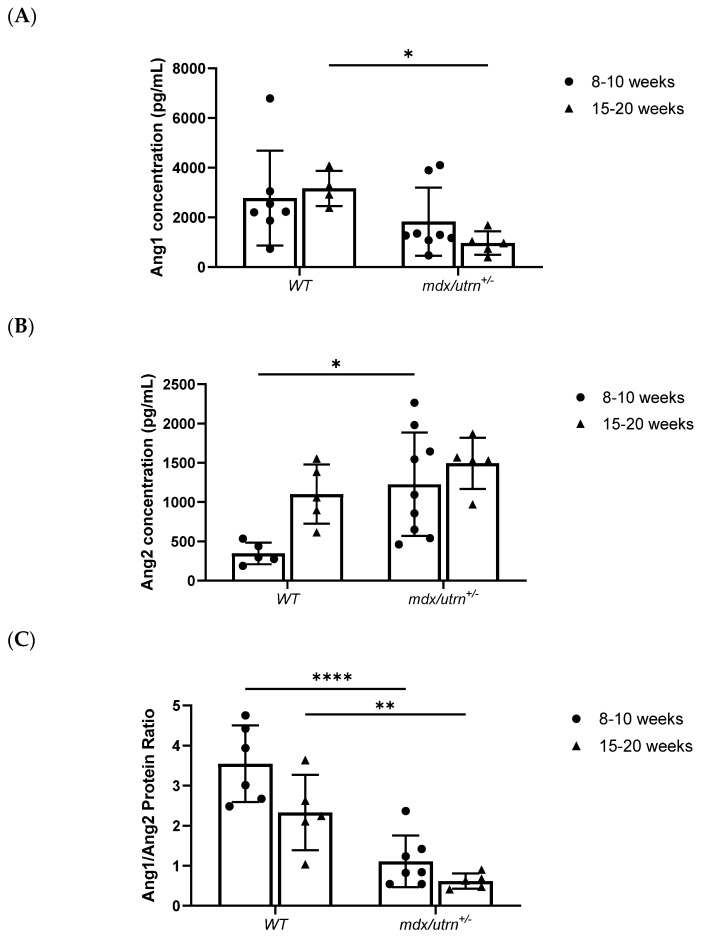
ELISA analysis of Ang1 and Ang2 protein expression. (**A**) ELISA quantification of Ang1 concentration in diaphragm lysates of young (8–10 weeks old) and mature (15–20 weeks old) WT and *mdx/utrn^+/−^* mice (*n* = 4–8). (**B**) ELISA quantification of Ang2 concentration in diaphragm lysates of young (8–10 weeks old) and mature (15–20 weeks old) WT and *mdx/utrn^+/−^* mice (*n* = 5–9). (**C**) Ang1 relative to Ang2 protein concentration in diaphragm lysates of young and mature WT and *mdx/utrn^+/−^* mice (*n* = 4–9). (**D**) Representative images of western blot of Ang1 and Ang2, and total-protein stain-free blots in diaphragm lysates of (8–10 weeks old) and mature (15–20 weeks old) WT and *mdx/utrn^+/−^* mice. Data are shown as individual values and mean ± SD. Statistics are performed using two-way ANOVA followed by Tukey’s test. * *p* < 0.05, ** *p* < 0.01, **** *p* < 0.0001.

**Figure 3 biomedicines-11-02265-f003:**
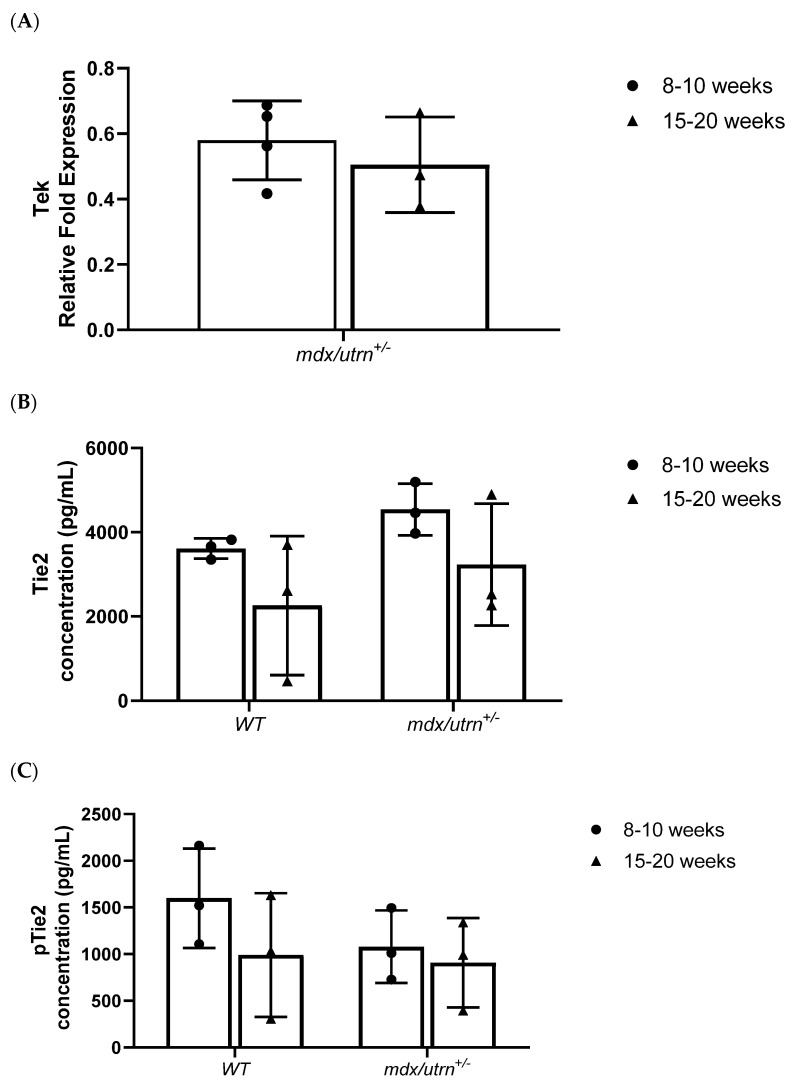
*Tek*, Tie2, and pTie2 expression. (**A**) RT-qPCR analysis of *Tek* (Tie2) relative fold expression in young (8–10 weeks old) and mature (15–20 weeks old) *mdx/utrn^+/−^* mice (*n* = 3–4) normalized to WT age–matched mice expression; reference genes: *ActB*, *AP3D1*, and *GusB*. (**B**) ELISA quantification of Tie2 concentrations in diaphragm lysates of young (8–10 weeks old) and mature (15–20 weeks old) WT and *mdx/utrn^+/−^* mice (*n* = 3). (**C**) ELISA quantification of pTie2 concentrations in diaphragm lysates of young (8–10 weeks old) and mature (15–20 weeks old) WT and *mdx/utrn^+/−^* mice (*n* = 3). (**D**) Ratio analysis of phosphorylated Tie2 (pTie2) relative to total Tie2 protein concentration in diaphragm lysates of young and mature *WT* and *mdx/utrn^+/−^* mice (*n* = 3). (**E**) ELISA quantification of VE–PTP protein concentration in young *WT* and *mdx/utrn^+/−^* mice (*n* = 3–6). Data are shown as individual values and mean ± SD. Statistics are performed using one-way ANOVA followed by Tukey’s test. * *p* < 0.05.

**Figure 4 biomedicines-11-02265-f004:**
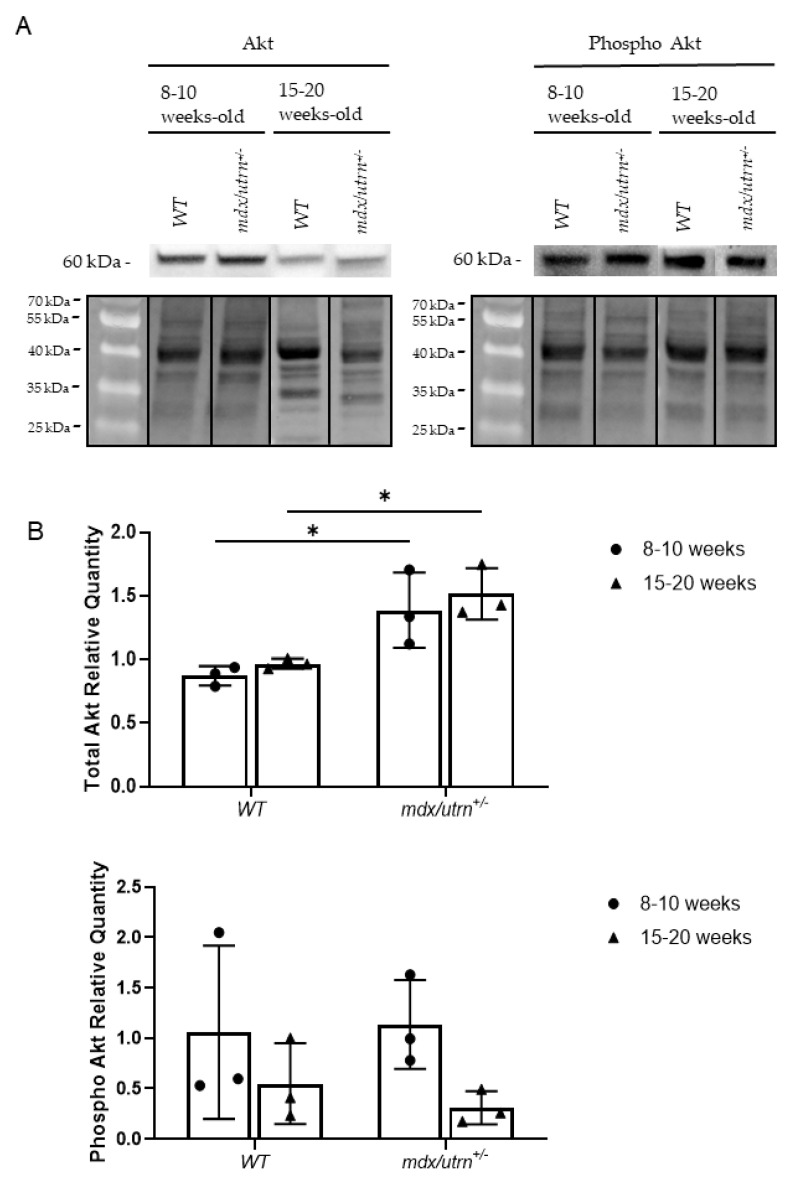
Western blot analysis of total Akt and phosphorylated Akt protein in diaphragm lysates. (**A**) Representative images of western blot of Akt and phosphorylated Akt (*Ser473*), and total–protein stain–free blots in diaphragm lysates of young and mature WT and *mdx/utrn^+/−^* mice. (**B**) Western blot semi-quantitative analysis of phosphorylated Akt and phosphorylated Akt in diaphragm lysates of young and mature WT and *mdx/utrn^+/−^* mice; normalized to total protein (*n* = 3). Data are shown as individual values and mean ± SD. Statistics are performed using two-way ANOVA followed by Tukey’s test. * *p* < 0.05.

**Figure 5 biomedicines-11-02265-f005:**
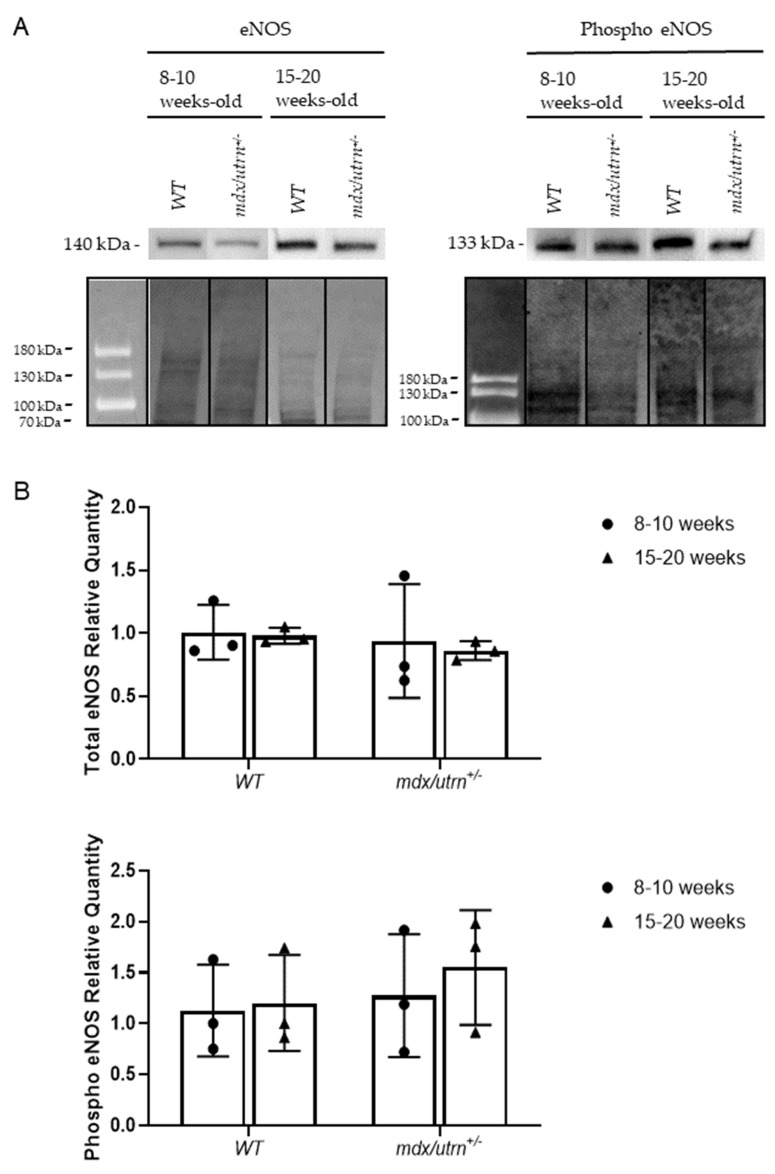
Western blot analysis of total eNOS and phosphorylated eNOS protein in diaphragm lysates. (**A**) Representative images of western blot of eNOS and phosphorylated eNOS, and total–protein stain–free blots in diaphragm lysates of young and mature WT and *mdx/utrn^+/−^* mice. (**B**) Semi-quantitative analysis of total eNOS and phosphorylated eNOS protein in diaphragm lysates of young and mature WT and *mdx/utrn^+/−^* mice; normalized to total protein (*n* = 3). Data are shown as individual values and mean ± SD. Statistics are performed using two-way ANOVA followed by Tukey’s test.

## Data Availability

Please contact the corresponding author for data inquiries.

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
