# Peer review of "Characterization of the Ang/Tie2 Signaling Pathway in the Diaphragm Muscle of DMD Mice"

_biomedicines, 2023, doi:10.3390/biomedicines11082265_

Round 1

Reviewer 1 Report (Previous Reviewer 3)

The revised manuscript has been improved and now it better clarify the focus of the study based on a characterization of  the  Ang1/Tie2 signaling pathway in diaphragm muscle of mdx mice.

Author Response

Thank you for reviewing our manuscript. We greatly appreciate your time and effort. 

Reviewer 2 Report (Previous Reviewer 4)

In the manuscript, the authors have shown the dysregulation of Ang1/Tie2 signalling in the mouse model of DMD. For this study, the authors have used the diaphragm of mdx/utrn+/- mice which is a better model for DMD. The methods are clearly described and the authors have discussed the drawback of the study. The manuscript is well written. The findings from this study are associative in nature which the authors have discussed. My comments are provided below

1.     In Fig1, the authors have shown the QRTPCR study. Please provide the primer details. The validation of expression should be done using two primer sets to correctly show the gene expression level. The authors should perform the QRTPCR using the primers targeting the 3’ end and 5’ end of gene.

2.     In Fig2 D, the authors should use the loading control either total protein level or GAPDH in addition to Ang2.

3.     In Fig4 A and 5A, the authors should use the loading control.

4.     The authors may consider removing the main findings section from the discussion section.

English language is fine with minor mistakes in spelling.

Round 2

Reviewer 2 Report (Previous Reviewer 4)

I have only one concern with the revised manuscript. The revised manuscript didnot have the images of the total protein level for the western blot analysis. The loading control should be presented in main figure.

There are minor spelling mistakes in the text.

Author Response

I have only one concern with the revised manuscript. The revised manuscript didnot have the images of the total protein level for the western blot analysis. The loading control should be presented in main figure.

Response: We have added the total protein levels to the main figures.

Round 3

Reviewer 2 Report (Previous Reviewer 4)

The authors have addressed all my concerns in the revised manuscript. I support the publication of the manuscript.

This manuscript is a resubmission of an earlier submission. The following is a list of the peer review reports and author responses from that submission.

Round 1

Reviewer 1 Report

This is an interesting article from a prestigious team. Some minor points are enlisted below"

1. Avoid abbreviations in the introduction section. 

2. Line 100~101: " reaching the desired age", please give a range of desired age. 

3. Please offer representative gel eletrophoresis genotyping results. If there is any reference (e.g., offered by the Jax), please specify the reference or offer the website linkage.   

4. Line 217: " despite the increase in mRNA concentration" , please try to rephrase the sentence, because the difference in figure 1 is Angpt1, not Angpt2. 

5. Line 255~263. If there p value > 0.05, just state that "there is no difference between".....No need to say " slight non-significant decrease".

6. Please mark the appropriate asterisk in figure 5, and cite figure 5 between 264~271. 

7. Figure 7A, the representative western figure showed that phosphorylated Akt seems to be higher in WT mice in 8-10 weeks old mice. However, the text described no difference (line 307-309)? 

8. line 361, explain the abbreviation COMP. 

I think that English is the native language for the author group. 

Reviewer 2 Report

In this paper, Lin and colleagues characterize angiopoietin (Ang) 1 and Ang2 levels and ratio and Ang receptor (Tie2) expression and activation (i.e. Tie2 phosphorylation and AKT activation) in a mouse model of Duchenne muscular dystrophy (DMD). Although the introduction and discussion sections are clear and well written, representing the main strength of this paper, results are very confusing and need a thoroughful revision to amend the inconsistencies between the text, the figures and figure legends. There is no clear connection linking the experiments, which are often poorly introduced, not allowing to understand what was the rationale that determined the author choice to perform them. Importantly, most of evidence provided is of correlative nature, and there are numerous flaws in the experimental approach. This do not allow any conclusions to be drawn with respect to the hypotheses formulated by the authors.

-It is not clear why authors decided to summarize main findings before starting to describe the results. This section should be moved to the discussion section.

- Results reported in lines 198-200 are not clear and it seems that there is no figure describing them. If it is so, please provide an appropriate figure to add in fig1 (which indeed currently consists of only one graph and, thus, is plenty of space), in a supplementary figure if preferred, or eliminate the paragraph as data not shown should  be avoided. Also please try to better describe the result and the experiments performed to obtain it.

- In fig1 please also report values and SD of controls ( i.e. WT animals), to help us to have a real idea of the result described in the text. This seems to be of importance, as p values reported in the text body and in the figure legend are different.

-Please, use letters to distinguish between the panels that compose a figure. For example, upper panel in figure2 should be named as fig 2a. Accordingly, in the text, the corresponding figure and panel should also be reported. This is one of the main point generating confusion along the paper.

-The authors should perform other experiments to confirm the results obtained in fig2. In particular, western blot could help to understand whether the results obtained are consistent with those observed by ELISA. Please review carefully what is reported in lines 210-217: it seems that some of the reported results are significant in the text but not in the figure, and there is confusion between the interpretation of mRNA and ELISA results. Note that the authors state that this result is interesting (line 203), but they do not discuss why, and more importantly, they provide no explanation, either in the results or in the discussion section.

-Why do authors report the ratio values in fig3? It would seem more logical, to put the difference of Angpt1/Angpt2 in fig1 results, while Ang1/Ang2 protein ratio in fig2. Notably, in the text, Ang1/Ang2 protein ratio is reported to be illustrated in fig4, which is not.

- fig5 and fig6 should be included in fig4.

- fig8 should be included in fig7

- western blot results (fig7 and 8) are not reliable. in addition to the fact that presenting results as the authors did raises questions, analysis of the whole blots submitted by the authors does not improves the situation. No molecular weights are reported. It definitely appears that the "n" for the evaluation of AKT, pAKT and eNOS levels is 2 and not 3, as reported in figure legend. Nevertheless, 3 samples are shown in the corresponding graphs. A numerous set of unknown bands accompanies that of the experiment shown in figure, which we certainly cannot say to be the most representative. Above all, it lacks a loading control, which is of utmost importance, as authors sustain that total AKT expression is modulated.

English is fine.

Reviewer 3 Report

Report

The absence of the cytoskeletal protein dystrophin represents the diagnostic marker of Duchenne muscular dystrophy (DMD).

In this manuscript, the authors aim to characterize the signaling pathway of angiopoietin 1 (Ang1)/endothelial cell receptor (Tie2) in a mouse model of Duchenne muscular dystrophy (DMD), in particular mdx/utrn+/- mice.

By different approaches, the authors provide evidence that Ang1/Tie2 signaling might be dysregulated in DMD.

Although very interesting and well-supported by methodologies, this study provides data only from the diaphragm isolated by mdx/utrn+/- mice.

This version of manuscript needs to provide further evidence of the dysregulation of Ang1/Tie2 signaling pathway by analysing at least an additional muscle, such as the tibialis anterior (TA), and extensor digitorum longus (EDL).

Reviewer 4 Report

In this manuscript, the authors have shown the Tie2 signaling pathway in the mouse model of DMD. The manuscript is well written. The methods are clearly described. the authors have used the better model of DMD. However, most of the results are preliminary in nature and they arenot supporting the conclusion. Finally, the study shows only the association of the Ang1/Ang2 to DMD. There are several demerits in the study which needs to be addressed before its final acceptance to publication. My comments are provided below

1. DMD is a progressive disease - fibrosis and vascular defects are during the later stage of disease. The authors have shown the result of the progressive changes in the level of Ang1 and Ang2 by western blot, QRTPCR in the whole tissue lysate. However, it needs to be shown in the level of tissue cross section (IHC or ISH).

2. The cryosections are better than the paraffin sections and the authors may consider it for their study. The authors didnot show any images of the tissue cross section.

3. Most of the study are associative in nature. Dystrophin is absent in MDX mice model and to compensate it many proteins including utrophin are upregulated in muscle. The authors have claimed that microenviornment of muscle also regulate the function in MDX mice but the mechanism is missing.

4. The authors should add a molecular weight markers in their western blot images.

Errors in English language are minor. 

Round 2

Reviewer 1 Report

My concerns have been properly addressed.

Reviewer 2 Report

I thank the authors for addressing some of the issues raised by this reviewer. Nevertheless, many of the most important issues remain unresolved and do not allow us to draw those conclusions that are essential to consider the authors' hypotheses as fulfilled. Importantly, the western blot data cannot be shown as the authors suggest, either in the figure that would be published or in the accompanying file they provided.

Reviewer 3 Report

In Duchenne muscular dystrophy (DMD), myofibers undergo to cycles of degeneration and regeneration. Muscle fiber types can mediate susceptibility or resistance to damage. Importantly, muscle fiber type impacts on molecular pathways underlying muscle diseases. Based on this critical issue, this study needs to also verify a muscle typically classified as fast-twitch muscles (such as EDL), in order to compare damage parameters observed in the diaphragm, and to provide a better characterization of the Ang1/Tie2 signaling pathway in DMD.    

Reviewer 4 Report

The authors didnot perform additional experiments as suggested by the reviewers neither did they change the conclusions of the manuscript. The image of the tissue cross section didnot help much as there is no statistics with it. The results arenot supporting the conclusion. The authors should at least discuss the weakness of the manuscript in the discussion section and tone down the claim.  

Quality of English language is fine.